# Isolation of Viable SARS-CoV-2 Virus from Feces of an Immunocompromised Patient Suggesting a Possible Fecal Mode of Transmission

**DOI:** 10.3390/jcm10122696

**Published:** 2021-06-18

**Authors:** Julie Dergham, Jeremy Delerce, Marielle Bedotto, Bernard La Scola, Valérie Moal

**Affiliations:** 1IHU-Méditerranée Infection, 13005 Marseille, France; julie.dergham.96@gmail.com (J.D.); jeremy.delerce@univ-amu.fr (J.D.); marielle.bedotto@gmail.com (M.B.); 2Institut de Recherche pour le Développement (IRD), Assistance Publique Hôpitaux de Marseille, Microbes Evolution Phylogeny and Infections (MEPHI), Aix Marseille Université, 13005 Marseille, France; 3Assistance Publique Hôpitaux de Marseille, Hôpital Conception, Centre de Néphrologie et Transplantation Rénale, 13005 Marseille, France

**Keywords:** SARS-CoV-2, stools, cell culture, Covid-19, organ transplantation

## Abstract

(1) Background: Severe Acute Respiratory Syndrome Coronavirus 2 (SARS-CoV-2) excretion in stools is well documented by RT-PCR, but evidences that stools contain infectious particles are scarce. (2) Methods: After observing a Corona Virus 2019 Disease (COVID-19) epidemic cluster associated with a ruptured sewage pipe, we search for such a viable SARS-CoV-2 particle in stool by inoculating 106 samples from 46 patients. (3) Results: We successfully obtained two isolates from a unique patient with kidney transplantation under immunosuppressive therapy who was admitted for severe diarrhea. (4) Conclusions: This report emphasizes that SARS-CoV-2 is an enteric virus, and infectious virus particles can be isolated from the stool of immune-compromised patients like, in our case, kidney transplant recipient. Immune-compromised patients are likely to have massive multiplication of the virus in the gastrointestinal tract and this report suggests possible fecal transmission of SARS-CoV-2.

## 1. Introduction

In Wuhan, China, a new viral disease emerged in December 2019. The pathogen responsible for this disease was identified as a new strain of coronavirus, called Severe Acute Respiratory Syndrome Corona Virus 2 (SARS-CoV-2), and the associated disease was named “Corona Virus 2019 Disease” (COVID-19) [1]. SARS-CoV-2 was responsible for more than 3.39 million deaths worldwide as of 19 May 2021 (https://covid19.who.int/ (accessed on 19 May 2021)). SARS-CoV-2 is highly contagious and considered as acquired through the respiratory tract after inhalation of particles or contact of face mucosa with contaminated hands [2]. For this reason, the main recommendations to avoid infection are the wearing of masks and frequent hand washing. However, mounting evidence suggests that this virus may also be an enteric virus. First, gastrointestinal symptoms have been described in patients with COVID-19 since the onset of the pandemic and a recent meta-analysis of >6600 patients with COVID-19 described that up to 15% had gastrointestinal symptoms, with the three most common symptoms being nausea or vomiting, diarrhea, and loss of appetite [3]. Second, the pooled estimate of SARS-CoV-2 viral RNA positivity in fecal samples was 54%, with positivity persisting for up to 47 days after symptom onset [3]. Apart from SARS-CoV-2 RNA detection, intracellular staining of viral nucleocapsid protein in gastric, duodenal, and rectal epithelia has been shown and demonstrated that SARS-CoV-2 infects these gastrointestinal epithelial cells [4]. Finally, we were able to observe that SARS-CoV-2 can grow in Caco-2 cells and polarized cells derivate from colorectal cancer [5]. In a recent work carried out in our institute, we identified a cluster of cases associated with a specific clone called genotype Marseille 1 [6]. The index case was imported from Tunisia, and the first cases subsequently diagnosed were associated with ships connecting North Africa to Marseilles, in travelers, but also in several crew members exposed to a ruptured sewage pipe. Travelers and crewmembers were infected with the same virus without direct contact with each other. We therefore raised the possibility of fecal-oral or fecal-respiratory transmission of SARS-CoV-2. Despite several attempts, viable SARS-CoV-2 was reported in stool of only six different patients [7,8,9,10]. In the present work, all SARS-CoV-2 PCR-positive samples from stools obtained in our laboratory were inoculated in order to evaluate the presence of a viable virus.

## 2. Materials and Methods

In this study, we attempted to isolate by cell culture SARS-CoV-2 from stool samples received in our laboratory at the IHU Méditerranée Infection for patients suffering from COVID-19. All samples were collected as part of the diagnosis and follow-up of patients for COVID-19, and the study was approved by the ethical committee of the University Hospital Institute Méditerranée Infection (N°: 2020-021). From 4 March 2020 to 29 April 2020, 128 stool samples (0.2 g in 1 mL of buffer, Sigma Virocult^®^, Elitech, Puteaux, France) from 54 patients were tested positive for SARS-CoV-2 by PCR targeting E gene [11]. Of these, 106 frozen samples from 46 patients and stored at −80 °C were available for viral isolation. This period of time was the starting month of the epidemic in Provence-Alpes-Côte d’Azur area, southeast of France. Stool samples were part of our initial protocol of sampling for all patients admitted at our center and hospitalized. After thawing, a 500 μL diluted sample was mixed with 150 μL of HBSS buffer and then filtered using a 0.22-μm pore-sized centrifugal filter (Merck Millipore, Darmstadt, Germany). Four wells of Vero E6 cells were each inoculated with 50 µL of the filtrate, as previously described [12]. The only change from the original protocol is that after the first week of subculture, instead of two blind subcultures each week, we performed five. It was abandoned due to its low yield and a massive influx of patients, which then limited our cultivation capacity. Once a cytopathic effect was detected in the well, the content of the well was collected. Six-hundred microliters were frozen to conserve the virus, and 200 μL was used to perform the SARS-CoV-2 qPCR for confirmation of presumptive identification, then whole genome sequencing [13]. The E gene of SARS-CoV-2 was amplified through RT-PCR (upstream primer: ACAGGTACGTTAATAGTTAATAGCGT; downstream primer: ATATTGCAGCAGTACGCACACA; probe: FAM-ACACTAGCCATCCTTACTGCGCTTCG-TAMRA). Sequencing was performed on Miseq Instrument with the Illumina Nextera XT Paired-end strategy. Genome consensus sequences were obtained by mapping reads with CLC Genomics workbench v7 against the genome Wuhan-Hu-1 (MN908947) with length fraction at 0.8 and similarity fraction at 0.9. The consensus sequence was analyzed with Nextclade web interface (https://clades.nextstrain.org/ (accessed on 24 May 2021) (version 0.7.5)).

## 3. Results

Four weeks after inoculation, i.e., at the third subculture, two samples showed cytopathic effects that appeared as a group of rounded cells forming aggregates comparing with the negative control, as shown in Figure 1. All other inoculations remained negative after the fifth sub-culture. RT-PCR performed on the two supernatants confirmed that the cytopathic effect was due to active SARS-CoV-2 proliferation, with cycle threshold (Ct) values of 17.29 and 16.22. Whole genome sequencing and analysis of isolate showed that the two strains had slightly different sequences of type 20A/8371T and 20B/19818T-28845T, respectively [14] (Figure 2). These two stool samples at a PCR Ct of 33.2 (2.585 copies/mL) and 33.4 (2.250 copies/mL), respectively with viable SARS-CoV-2, were collected on 14 and 15 April 2020 from the same patient. This patient was a 62-year-old man who had undergone a kidney transplant 21 years ago. He also had diabetes, hypertension, and was overweight. He consulted in the emergency department on 13 April because for the past 10 days he had been experiencing asthenia, loss of appetite, diarrhea, and weight loss, without respiratory symptoms (Figure 3). COVID-19 pneumonia was diagnosed on chest CT. SARS-CoV-2 PCR performed on nasopharyngeal swab two times per day on 13 and 14 April was negative. It was positive once on nasopharyngeal swab on 15 April at a Ct of 33.5 (2.099 copies/mL). The culture was negative for this swab, but direct amplification and sequencing on the sample allowed us to determine it was a 20A/8371T sequence type as in the first stool sample. Laboratory results on the day of admission revealed acute kidney injury and mild inflammation. Maintenance immunosuppressive treatment consisted of tacrolimus 6.5 mg/day and prednisone 5 mg/day. Treatment with azithromycin was administered for 5 days, hydroxychloroquine for 10 days and ceftriaxone for 7 days from 14 April. The dose of tacrolimus was temporarily halved. Acute kidney injury was due to dehydration following severe diarrhea and was corrected by intravascular fluids expansion and discontinuation of diuretics and ACE inhibitors. C reactive protein normalized on 18 April. Nasopharyngeal SARS-CoV-2 PCR was positive only once on 15 April and was negative on 21, 22 and 28 April. We did not test stool samples using SARS-CoV-2 PCR until 28 April, and the result was negative.

## 4. Discussion

In this work, we sought to determine whether the SARS-CoV-2 RNA positive stool contained infectious virus and then whether the stool could be a source of transmission. We succeeded in isolating viable SARS-CoV-2 only in 2/106 (1.9%) stool samples from 1/46 (2.2%) patients with COVID-19. To date, viable cases of SARS-CoV-2 have been reported in the feces of only six different patients, despite the common detection of viral RNA [7,8,9,10]. All of these patients were Chinese and contracted COVID-19 during the first trimester of 2020. One patient developed diarrhea [10], while two others did not [7]. No information about gastrointestinal symptoms was available for the last three patients. The viral load was considered high in two patients, but copy number was not provided [7]; to be at Ct between 20 and 24 in two patients [8]; and at 33.6 in 1 patient [10], as in our case. It was not indicated for the sixth patient [9].

Despite relatively high Ct, we succeeded in isolating viable SARS-CoV-2 in two different stools sampled 1 day apart from the same patient. COVID-19 presented in this patient as enteric infection, evolving for 10 days at the time of diagnosis and severe enough to cause acute kidney injury. Although pneumonia was detected by CT scan, respiratory symptoms were absent throughout the illness. This observation suggests that gastrointestinal infections can occur before respiratory symptoms [15,16], but also without them. Interestingly, our patient also had low viral excretion in the upper respiratory sample. The fact that two culture-positive stool samples have a viral load comparable to that of culture-negative nasopharyngeal sample suggests that there was continued viral multiplication in the digestive tract of this patient. Several studies suggested that SARS-CoV-2 may be actively replicating in the gastrointestinal tract [10], even after viral clearance in the respiratory tract [17]. The viral excretion from the digestive tract may last longer than that from the respiratory tract, since fecal samples may remain positive for SARS-CoV-2 RNA for approximately 5 weeks after respiratory tract samples become negative for SARS-CoV-2 RNA [17]. 

We cannot exclude the possibility that viral load in the stool could be high but appear low due to PCR inhibitors present in stools. However, as isolation occurred at the third subculture, the viral load was likely low. We identified two different SARS-CoV-2 strains in two stools sampled 1 day apart from the same patient. Genome sequence differences were probably not related to subcultures as the comparison of more than 50 SARS-CoV-2 full-length genomes sequenced in our laboratory showed no significant differences between the sequences from isolates or samples. Genome sequences of the two strains were sufficiently different to hypothesize a dual SARS-CoV-2 infection as it has already been reported [18,19], rather than genome variants related to ongoing evolution of SARS-CoV-2 in the human body [20].

We believe that isolation of viable SARS-CoV-2 only in stools of an immunocompromised kidney transplant recipient in this work is not by chance. In a large French nationwide cohort of 279 kidney transplant recipients with COVID-19, diarrhea was the third most frequent symptom on admission (43.5%) after fever (80%) and cough (63.6%) [21]. Gastrointestinal symptoms were significantly more frequent than those reported in the general population [3]. It remains uncertain whether the immunosuppression state contributes to the high proportion of gastrointestinal signs in the kidney transplant recipients, but immunosuppressants might favor high viral load and more active replication in gastrointestinal tract leading to more gastrointestinal symptoms. Higher COVID-19-related mortality compared to non-transplant hospitalized patients was reported despite a similar occurrence of severe disease [22]. SARS-CoV-2 plasma load was reported to be associated with COVID-19 severity and mortality, and respiratory shedding to be prolonged [22]. Moreover, patients receiving profound immunosuppression following hematopoietic stem-cell transplantation or receiving cellular therapies may excrete viable SARS-CoV-2 for at least 2 months in respiratory samples [23]. As in other viral infections in kidney transplant recipients, SARS-CoV-2 will probably more fully display its potential dangerousness than immunocompetence [24].

The number of times SARS-CoV-2 may have developed in the stool is very low. Performing cell culture with cytotoxic fecal specimens is technically challenging. In a study from Germany, virus isolation from stool samples was never successful, irrespective of viral RNA concentrations, in 13 samples from four patients [25]. We believe that our procedure using filtered diluted specimens without the addition of any potentially toxic antibacterial agents and cell culture medium changing after centrifugation is responsible for high virus recovery [12,26]. As this virus was totally new for us and we had no idea if it was easy or difficult to isolate, we did five subcultures, a technique traditionally used to improve culture yield. We indeed observed that two subcultures improved their percentage of isolated viruses [26]. The yield obtained by three additional subcultures, even if it was not zero, was not continued beyond the first 2 months of the epidemic at our center and is now reserved in case we try to isolate at all costs the viral strain despite relatively high Ct (low viral loads); typically at this time some people are infected despite a complete vaccination protocol. The present work and others have shown that the virus can survive in the digestive tract [7,8,9,10]. Jeong et al., although they failed to directly demonstrate the presence of viable virus in stools using cell culture isolation, were able to isolate SARS-CoV-2 from ferrets that were inoculated with a stool sample from a COVID-19 patient [27]. 

In conclusion, infectious SARS-CoV-2 particles can be shed in stools of COVID-19 patients. SARS-CoV-2 is an enteric virus, and SARS-CoV and Middle Eastern Respiratory Syndrome (MERS-CoV) can be transmitted through fecal-oral route. Immune-compromised patients are more likely to have high viral replication in the gastrointestinal tract, leading to more gastrointestinal symptoms, and are more likely to shed infectious virus in the stool than immune-competent patients. We suggest adding SARS-CoV-2 to the list of viruses to systematically look for in stool in immunocompromised patients in the case of diarrhea.

## Figures and Tables

**Figure 1 jcm-10-02696-f001:**
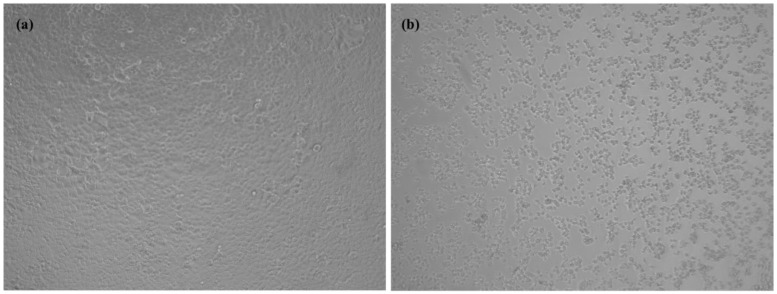
Cytopathic effect of Severe Acute Respiratory Syndrome Corona Virus 2 (SARS-CoV-2) on Vero E6 cells: (**a**) uninfected cells as negative control and (**b**) infected cells with the stool samples. The images were captured simultaneously at 4 days post-infection at sub-culture 3 using ZEISS Zen Microsoft software with ×10 magnification scale.

**Figure 2 jcm-10-02696-f002:**
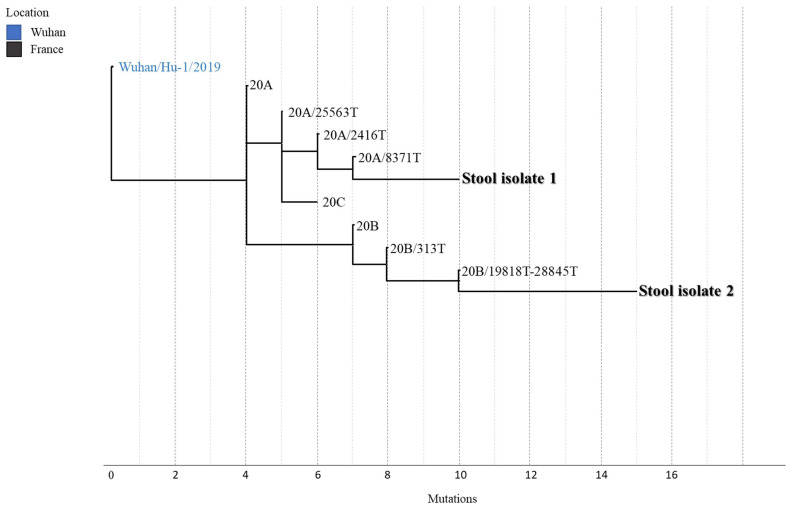
Phylogenetic tree showing the positions of the two SARS-CoV-2 strains isolated from stools relative to other phylogenetically close neighbors. Stool isolate 1 and 2 represent the two isolated strains from clinical samples of a kidney transplant patient. Nomenclature was based on Nextstrain. Genomic sequences of isolates 1 and 2 are available on GISAID under accession numbers EPI_ISL_860093 and EPI_ISL_860094, respectively.

**Figure 3 jcm-10-02696-f003:**
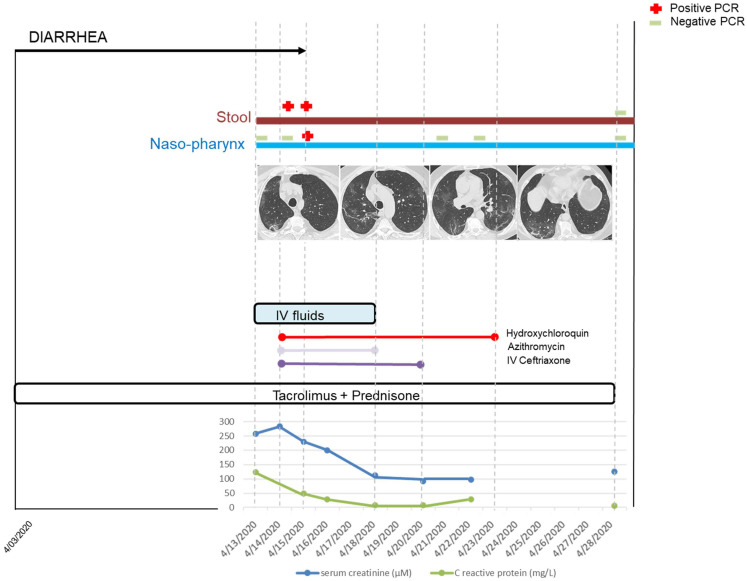
Clinical, biological, virological, and treatment timeline during the course of Corona Virus 2019 Disease (COVID-19). Gastroenteritis began 10 days before hospitalization. Diarrhea ceased on 15 April. SARS-CoV-2 PCR was positive first in the stool and then in the pharynx. Typical COVID-19 pneumonia existed on the CT-scan, while the patient presented no respiratory symptom. Acute kidney injury was due to dehydration following severe diarrhea and was corrected by intravascular fluids expansion and discontinuation of diuretics and ACE inhibitors. The dose of tacrolimus was temporarily halved. Treatment with azithromycin was administered for 5 days, hydroxychloroquine for 10 days, and ceftriaxone for 7 days. C reactive protein normalized on 18 April. The two consecutive fecal samples were positive for SARS-CoV-2 by RT-PCR and culture. IV: intravenous.

## Data Availability

Genomic sequences of stool isolates 1 and 2 are available on GISAID under accession numbers EPI_ISL_860093 and EPI_ISL_860094, respectively.

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
