# Peer review of "Isolation of Viable SARS-CoV-2 Virus from Feces of an Immunocompromised Patient Suggesting a Possible Fecal Mode of Transmission"

_jcm, 2021, doi:10.3390/jcm10122696_

Round 1

Reviewer 1 Report

In this work, Authors tried to demonstrate the presence of viable and infectious virus in SARS-CoV-2 PCR-positive samples of stools by inoculating diluted sample in Vero E6 cells. They succeeded in isolating viable SARS-CoV-2 in 2 out of 106 stool samples  from 1 out of 46 patients with COVID-19. 

The isolation of infectious SARSCoV-2 viruses from stool samples of COVID-19 patients has already been tried in three Chinese patients. Further, a viable SARS-CoV-2 was isolated from nasal washes of ferrets inoculated with patient stool. This work is a further demonstration that SARS-CoV-2 is also an enteric virus.  Instead, in a study from Germany, virus isolation from stool samples was never successful, irrespective of viral RNA concentrations, in 13 samples from four patients [Wölfel R, Corman VM, Guggemos W, Seilmaier M, Zange S, Müller MA, Niemeyer D, Jones TC, Vollmar P, Rothe C, Hoelscher M, Bleicker T, Brünink S, Schneider J, Ehmann R, Zwirglmaier K, Drosten C, Wendtner C (2020) Virological assessment of hospitalized patients with COVID-2019. Nature 581:465-469].  The experience of the German work should be mentioned also in support of the fact that, due to the cytotoxicity of faecal specimen, performing cell culture with stool is technically challenging.

Understanding the various ways of transmission of the virus is important to contain  its rapid spread.

Author Response

Dear Reviewer 1, 

Our manuscript has been thoroughly amended in accordance with your advices. Please find in the attachment our item by item responses to each point brought up, inserted underneath each comment, for our revised manuscript newly entitled “Isolation of viable SARS-CoV-2 virus from Feces of an immunocompromised patient suggesting a possible fecal mode of transmission”. 

We are thankful to the reviewers for helping us to improve our manuscript and hope that you will find this revised article suitable for publication in the Journal of Clinical Medicine.

Kindly regards,

Pr. Valérie Moal

Reviewer 2 Report

The authors have done a remarkable job in completing this study. Here are few suggestions 

Title:

Isolation of Viable SARS-CoV-2 in Feces Suggesting Possible 2 Fecal Transmission of COVID-19

Since the authors have only isolated the viable virus in one patient with a kidney transplant and the patient was on immunosuppression, this title is misleading. The title suggests that this is a critical route of transmission, which is not the case as less than 2 percent of their patient population have a viable virus.

Please change the title to a more appropriate one  like

“Feces as the mode of transmission for SARS-CoV-2 in patients with immunosuppression”

Or

“Isolation of viable SARS-CoV-2 virus from Feces of an immunocompromised patient suggesting a possible fecal mode of transmission”

The second title is more appropriate for search engine optimization.

Abstract:

Line  15 conclusion :

This report emphasizes that SARS-CoV-2 is 15 also an enteric virus that is important to detect in the stool in cases of diarrhea, particularly after 16 kidney transplantation

Suggestion:  This report emphasizes that SARS-CoV-2 is an enteric virus, and infectious virus particles can be isolated from the stool of immunosuppressed patients like, in our case, kidney transplant patient.  

Introduction

Line 26   "COronaVIrus 2019 Disease” (COVID-19)

Suggestion:  correct it to Corona Virus – 2019

Line 28    SARS-CoV-2 is highly contagious and considered as ac- 28 quired through the respiratory tract after inhalation of particles or contact of face 29 mucosae with contaminated hands

  • Provide a reference for it

Line 38  - SARS-CoV-2  RNA detection, but also intracellular staining of viral nucleocapsid protein in gastric, duodenal and rectal epithelia demonstrated that SARS-CoV-2 infects these gastrointestinal epithelial cells

  • This line doesn't make sense. It seems like Author wants to say, “Apart from SARS-CoV-2 RNA detection, studies have shown intracellular staining of viral nucleocapsid protein in gastric, duodenal and rectal epithelia demonstrated that SARS-CoV-2 infects these gastrointestinal epithelial cells.”

Material and methods:

Please describe which patients were selected for stool samples?  All patients admitted in this time range or only patient who has GI symptoms?

Why did Author decide to do five subcultures? It will be nice if they can explain their rationale for this

Results:

103 - Acute functional renal failure secondary to diarrhea corrected after refilling and discontinuing diuretics and ACE inhibitors

  • What does the Author mean by Acute functional renal failure?
  • What does the Author mean by “ after refilling”

Line 106  What does the Author means by “It has 106 not been controlled since April 15, when diarrhea stopped ?”

Also seem like from Figure 3 , RT- PCR from stool was positive earlier. The Author might want to mention it in the description in the results section.

Did the Author do stool culture on April 28 sample? I am guessing it was -ve . Seem like at least PCR was -ve so what this stool was cultured ? or as it not cultured? 

Discussion : 

Line 148 : The viral excretion from the digestive tract may last longer than that from the respiratory tract since fecal samples may remain positive for SARS-CoV-2  RNA for approximately five weeks after respiratory tract samples become negative for  SARS-CoV-2 RNA,

Suggestion: this can be added:    though in our case also the repeat stool sample culture on 28 was -ve ( if it was done ) suggesting that like upper respiratory tract,  viral shedding can be detected by PCR much longer than infectious viral shedding in GI tract.  If the culture was not done, then this statement can not be added .

Line 167 : It remains uncertain whether the immunosuppression state contributes to the high proportion of gastrointestinal signs in the kidney transplant recipients

Suggestion:  this can be added after this lien if the Author agrees ?: - “but immunosuppressant might cause high viral load and more active replication in GI tract leading to more GI symptoms”

Line 186 : In this statement, is the Author suggesting that stool culture or PCR should be done in every kidney transplant patient?

Suggestion : This can be rephrased as 

In conclusion, SARS-CoV-2 is also an enteric virus, and infectious viral particles can be shed in stool as in our case and can cause infection by the oral fecal route. Immunocompromised patients are more likely to have high viral replication in GI tract leading to more GI symptoms, and more likely to shed infectious virus in the stool than immunocompetent patients. Further studies are needed in this area.

references : 

1 . this will be a better reference than the quoted reference 

Naming the coronavirus disease (COVID-19) and the virus that causes it (who.int)

Author Response

Dear Reviewer 2,

Our manuscript has been thoroughly amended in accordance with your requests, advices and comments. Please find in the attachment our item by item responses to each point brought up, inserted underneath each comment, for our revised manuscript newly entitled “Isolation of viable SARS-CoV-2 virus from Feces of an immunocompromised patient suggesting a possible fecal mode of transmission”.

We are thankful to the reviewers for helping us to improve our manuscript and hope that you will find this revised article suitable for publication in the Journal of Clinical Medicine.

Kindly regards,

Pr. Valérie Moal

Round 2

Reviewer 2 Report

Dear Author thanks for your reply. Will be recommending the research for publication.